# Spreading Allergic Contact Dermatitis to Tea Tree Oil in an Over-the-Counter Product Applied on a Wart

**DOI:** 10.3390/medicina58050561

**Published:** 2022-04-19

**Authors:** Francesca Ambrogio, Caterina Foti, Gerardo Cazzato, Edoardo Mortato, Stella Mazzoccoli, Anna Paola De Caro, Nicoletta Cassano, Gino Antonio Vena, Gianfranco Calogiuri, Paolo Romita

**Affiliations:** 1Section of Dermatology, Department of Biomedical Science and Human Oncology, University of Bari, 70124 Bari, Italy; dottambrogiofrancesca@gmail.com (F.A.); edoardo.mortato@gmail.com (E.M.); stellamazzoccolimd@gmail.com (S.M.); annapaola.decaro@gmail.com (A.P.D.C.); paolo.romita@uniba.it (P.R.); 2Section of Pathology, Department of Emergency and Organ Transplantation (DETO), University of Bari Aldo Moro, 70124 Bari, Italy; gerardocazzato8@gmail.com; 3Dermatology and Venereology Private Practice, 70100 Bari, Italy; nicoletta.cassano@yahoo.com (N.C.); ginovena@gmail.com (G.A.V.); 4Dermatology and Venereology Private Practice, 76121 Barletta, Italy; 5Hospital Vito Fazzi, 73100 Lecce, Italy; gf.calogiuri@libero.it

**Keywords:** allergic contact dermatitis, *Melaleuca alternifolia*, tea tree oil, viral warts, plantar warts, topical treatment

## Abstract

Tea tree oil is an essential oil obtained by distillation from the leaves and terminal branchlets of *Melaleuca alternifolia* and is now present in numerous products for body care and self-medication. We report a case of allergic contact dermatitis to tea tree oil in a young man who was applying a lotion containing tea tree oil on a wart localized on the plantar aspect of the right big toe, which had previously been treated with cryotherapy. He developed a severe eczematous eruption on the right foot and the right leg, with subsequent id reactions affecting the right thigh, the contralateral lower limb, the trunk and the upper limbs. The lotion was discontinued, and the dermatitis resolved after topical corticosteroid therapy. Patch testing with the aforementioned lotion 10% pet. and oxidized tea tree oil 5% pet. identified tea tree oil as the culprit agent of the dermatitis. This case report confirms that products made of natural ingredients, often perceived to be harmless, can cause allergic reactions.

## 1. Introduction

Cutaneous warts are common skin conditions caused by some strains of human papilloma virus [1]. Plantar warts can be painful and disabling, limiting ambulation and daily activities, and their management is often challenging because of the risk of treatment failure and recurrence. The list of potential treatments for cutaneous warts is wide. Such treatments are not uniformly successful, with varying cure rates and overall low-quality evidence [2,3]. Keratolytic agents and destructive treatments, in particular salicylic acid and cryotherapy, are included among the first-line therapeutic approaches [4,5].

Natural, complementary and alternative remedies are used for medical purposes, including treatment of warts and other cutaneous conditions, and, among natural products, essential oils are increasingly popular [6,7,8].

Tea tree oil (TTO) is the volatile essential oil derived mainly from *Melaleuca alternifolia,* an Australian native plant belonging to the family of *Myrtaceae*. The composition of TTO is highly complex. In fact, more than 100 components have been identified, most of which are monoterpenes and related alcohols, and the concentration of such constituents can vary depending on the sample [9,10].

TTO is incorporated in many topical preparations because of its antimicrobial, antioxidant and anti-inflammatory activities and is regarded as a natural remedy for various skin diseases and in aromatherapy [7,9,11,12,13]. Various applications of TTO in dermatology have been proposed, such as treatment of acne, seborrheic dermatitis, insect bites, sunburn, wounds, warts, herpes simplex and fungal infections [7,9,10,12]. Nonetheless, contact dermatitis, even severe, may result from the application of TTO as pure oil and also diluted in cosmetics and over-the-counter preparations, as well as from occupational exposure [10,14].

## 2. Case Report

A 28-year-old non atopic male presented to our department with an itchy and widespread eczematous eruption that had begun 7 days earlier, initially on the right foot and then on the right leg. In these two areas the dermatitis became particularly severe (Figure 1). In the following days, id reactions involved his right thigh and the contralateral lower limb, and then a few discrete eczematous lesions appeared on his trunk and upper limbs. The patient was applying an over-the-counter lotion containing TTO, salicylic acid and lactic acid on a wart localized on the plantar aspect of the right big toe (Figure 2), which had been treated with cryotherapy one month earlier. Eight days after the cryotherapy session, because of the persistence of the wart and related pain, the patient started the use of the over-the-counter lotion that was applied once daily on the damaged skin for nearly 20 days without seeking a medical advice. The dermatitis resolved after treatment with clobetasol propionate cream applied twice a day for 10 days and the discontinuation of the lotion.

Three weeks after clinical remission the patient underwent patch testing with the SIDAPA (Società Italiana Dermatite Allergologica, Professionale ed Ambientale) standard series (Euromedical, Calolziocorte, LC, Italy), plus the aforementioned lotion 10% pet. and oxidized TTO 5% pet. (Chemotechnique MB Diagnostics AB, Vellinge, Sweden). Patch tests were carried out at the Dermatology Clinic of University of Bari using Al-test^®^ (Euromedical, Calolziocorte, LC, Italy). Readings were performed at day (D) 2 (after 48 h), D4 (after 96 h), and D7 (after approximately 168 h) according to SIDAPA guidelines [15]. Patch tests showed a positive reaction (++) at D2 and D4 for colophony, the over-the-counter preparation (Figure 3) and TTO (Figure 4). In the patient’s history, there were no previous episodes of eczema and no reactions following the exposure to potential colophony sources. The patient denied the use of any TTO-containing topical preparations in the past. Furthermore, patch tests with the lotion 10% pet. and oxidized TTO 5% pet. were also performed in a control group of 10 healthy subjects with negative results.

A diagnosis of allergic contact dermatitis (ACD) to TTO was made and the patient was recommended to avoid products containing *Melaleuca* extracts without any relapse of the dermatitis throughout a 3-month follow-up period.

## 3. Discussion

TTO (CAS 68647-73-4) is an essential oil obtained by steam distillation of the leaves and terminal branchlets of *Melaleuca alternifolia* (tea tree). TTO is characterized by anti-oxidant, anti-inflammatory and broad-spectrum anti-microbial properties and has been used for many decades to treat various skin disorders [9,10]. It is nowadays present in several topical products for body care and self-medication.

Nevertheless, topical use of TTO has been associated with adverse effects, such as skin irritation and ACD [10]. Immediate systemic hypersensitivity has rarely been described [16]. Systemic contact dermatitis after oral administration and airborne ACD caused by inhalation of a TTO aqueous solution have also been reported [17,18]. The literature contains numerous reports of ACD to TTO showing cases with lesions limited at the site of application or spreading, occasionally all over the body [10,19,20]. ACD with an extensive erythema multiforme-like reaction has been observed [21], as well as linear IgA disease precipitated by contact dermatitis to TTO [22]. Taking into account the widespread use of TTO-based products, it is possible that contact allergy to TTO is underestimated because this allergen is present only in supplementary series of patch tests that are not always used [15].

Attempts have been made to identify the exact allergens present in TTO. The most frequent sensitizers in TTO appear to be ascaridole, terpinolene, alpha-terpinene trihydroxymenthane, alpha-phellandrene and limonene [10]. Fresh TTO was revealed to be a weak sensitizer, whereas the sensitizing potential was found to be increased by oxidation [23]. In fact, oxidation products have been recognized as relevant allergens [23]. Oxidation may be enhanced by particular conditions, such as long storage and frequent opening of bottles, as well as exposure to light, heat, humidity and oxygen [7]. Skin sensitization may also be amplified by irritancy [10].

Many cases of ACD to TTO were caused by the use of undiluted oil or products with high concentrations, often applied on damaged skin [10]. In this respect, some authors have warned against the use of TTO in concentrated forms, particularly on damaged skin [14]. Therefore, it is important to avoid the exposure to aged (oxidized) oils and highly concentrated forms in order to reduce the risk of sensitization to TTO [10]. For the same reason, it is important to perform patch tests with oxidized TTO, as well as with the culprit product [20]. In 2002, the European Cosmetic Toiletry and Perfumery Association (COLIPA) recommended that TTO should not be used in cosmetic products in a way that results in a concentration greater than 1% oil being applied to the skin. Furthermore, when formulating TTO in a cosmetic product, manufacturers were encouraged to consider the use of antioxidants and/or specific packaging to minimize exposure to light in order to reduce the formation of oxidation products [24].

We observed a severe and disseminated reaction probably related to the use of the sensitizing product on damaged skin that might have promoted sensitization to TTO and the spread of skin reaction. We cannot rule out that the concomitant presence of keratolytic agents in the over-the-counter preparation could have affected the absorption of TTO. TTO concentration might be another factor influencing the development and severity of allergic reactions. In the literature, less severe reactions were usually reported with exposure of the intact skin to low TTO concentrations [10]. Unfortunately, the concentration of TTO in the lotion was not known.

A concomitant contact sensitization to colophony was observed in our patient, who had a negative history of reactions to band aids, medical tapes or other potential sources of colophony. ACD to TTO may be associated with sensitization to other allergens, such as oil of turpentine, fragrance mix and other essential oils [10]. In various publications, ACD to TTO has been described in association with sensitization to colophony [14,18,21,22,23,25,26,27]. However, because of the complexity of TTO-containing compounds, it is difficult to establish whether such positive reactions are attributable to concomitant sensitization, cross-reactivity, or pseudo cross-reactivity (common allergenic ingredients).

## 4. Conclusions

Our report contributes to increase the number of cases of ACD to TTO and confirms that products made of natural ingredients, often considered harmless, can cause allergic reactions.

Further studies are necessary to determine the exact sensitizing substances present in TTO trying to define also the most appropriate concentrations for their testing.

## Figures and Tables

**Figure 1 medicina-58-00561-f001:**
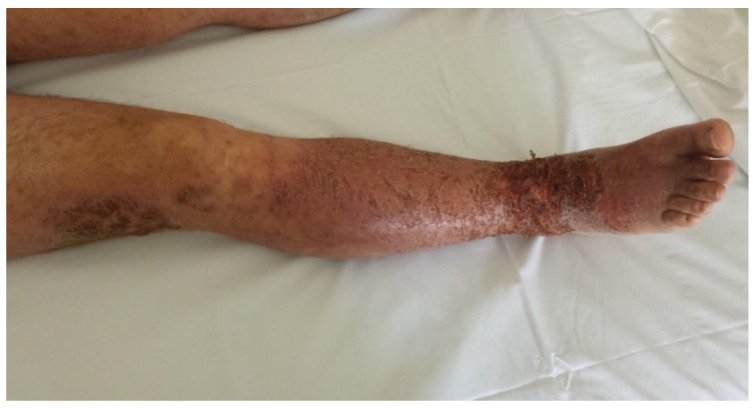
Disseminated eczematous reaction after the use of an anti-wart product containing tea tree oil.

**Figure 2 medicina-58-00561-f002:**
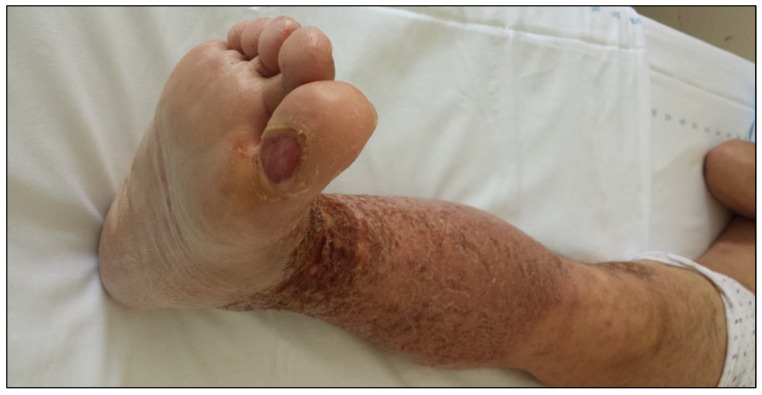
The site of application of the lotion containing tea tree oil.

**Figure 3 medicina-58-00561-f003:**
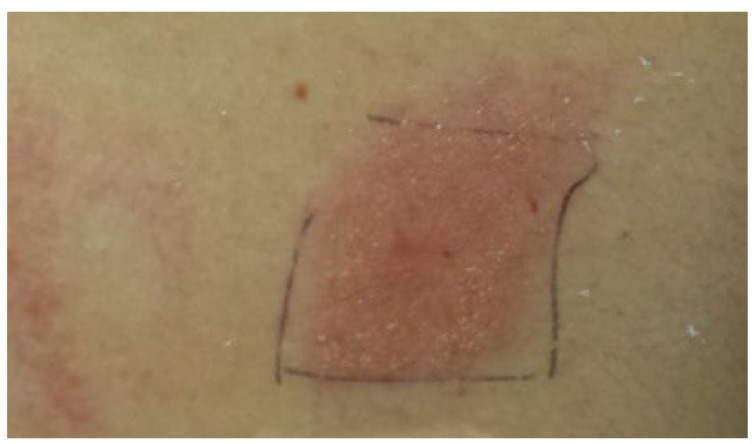
Patch test with the lotion 10% pet.

**Figure 4 medicina-58-00561-f004:**
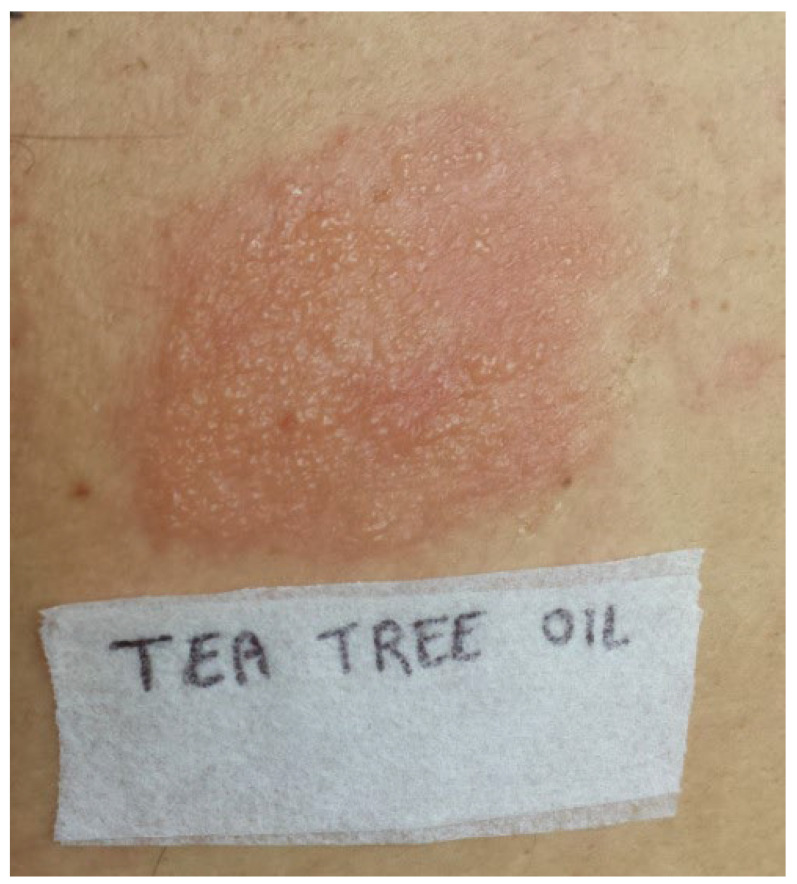
Patch test with oxidized tea tree oil 5% pet.

## Data Availability

Not applicable.

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
