# Peer review of "Spreading Allergic Contact Dermatitis to Tea Tree Oil in an Over-the-Counter Product Applied on a Wart"

_medicina, 2022, doi:10.3390/medicina58050561_

Round 1

Reviewer 1 Report

The manuscript „Spreading allergic contact dermatitis to tea tree oil in an over-the-counter product applied on a wart“ presents the case report, based on the author` s experience with the patient manifested with contact dermatitis to tea tree oil.

  This manuscript is valuable and may contribute to the general knowledge on this subject and further work with similar clinical pictures and cases. In addition, I consider the figures very useful for the manuscript and its significance for further management of patients.

However, I have some suggestions:

- ABSTRACT: Please add information on the type of skin lesions and final data on regression of the lesions after therapy and remove tree oil (to the Abstract), as they could be important for readers.

- Please mention the numbers of hours (48, 72, etc.) of the patch test reactions readings (I am not sure that all readers know their meanings). Also, please mention where the test was conducted (Was it in a private dermatology clinic or at the Department of dermatology?).

- The text of the manuscript is somewhat interrupted and I suggest you form the paragraphs according to the specific subject of each paragraph. For instance, some sentences are disconnected from paragraphs and are alone.   

-CONCLUSION: The sentence „Moreover, taking into account the widespread…“ is  possible author's explanation and please move this sentence to the  Discussion section.

Finally, I suggest checking the text by an English professional or a native speaker.

Author Response

Referee n. 1

The manuscript „Spreading allergic contact dermatitis to tea tree oil in an over-the-counter product applied on a wart“ presents the case report, based on the author` s experience with the patient manifested with contact dermatitis to tea tree oil. This manuscript is valuable and may contribute to the general knowledge on this subject and further work with similar clinical pictures and cases. In addition, I consider the figures very useful for the manuscript and its significance for further management of patients.

Reply: Thank you very much

However, I have some suggestions:

- ABSTRACT: Please add information on the type of skin lesions and final data on regression of the lesions after therapy and remove tree oil (to the Abstract), as they could be important for readers.

Reply: You are right. This information was added.

- Please mention the numbers of hours (48, 72, etc.) of the patch test reactions readings (I am not sure that all readers know their meanings). Also, please mention where the test was conducted (Was it in a private dermatology clinic or at the Department of dermatology?).

Reply: Such details were specified.

- The text of the manuscript is somewhat interrupted and I suggest you form the paragraphs according to the specific subject of each paragraph. For instance, some sentences are disconnected from paragraphs and are alone.  

Reply: We tried to improve this aspect.

-CONCLUSION: The sentence „Moreover, taking into account the widespread…“ is  possible author's explanation and please move this sentence to the  Discussion section.

Reply: The sentence was placed in the Discussion section.

Finally, I suggest checking the text by an English professional or a native speaker.

Reply: English grammar and spelling were checked

Reviewer 2 Report

Even the authors claim "The literature contains numerous reports" in which they are now presenting one single case.

This one case alone provides neither sound, nor new, relevant message. Based on that I recommend: reject.

Author Response

Referee n. 2

Even the authors claim "The literature contains numerous reports" in which they are now presenting one single case. This one case alone provides neither sound, nor new, relevant message. Based on that I recommend: reject.

Reply: There are numerous reports documenting the possible occurrence of contact allergy to tea tree oil. However, we think it is important to report similar cases in order to increase our knowledge and to underline that natural products are not always safe. Moreover, our report also shows that allergic contact dermatitis to tea tree oil can be severe and widespread. 

Reviewer 3 Report

The paper "Spreading allergic contact dermatitis to tea tree oil in an over-the-counter product applied on a wart" is an interesting clinical case, which shows that treatment with over-the-counter products can be harmful. Patients often abuse over-the-counter products, especially natural ones, without fear of side effects. All side effects of such products should be reported and recorded. The paper contains relevant knowledge, especially for primary care physicians who may have patients with such complications. It is important to distinguish between allergic and irritant reactions. All these aspects were discussed in the article.

Author Response

Referee n. 3

The paper "Spreading allergic contact dermatitis to tea tree oil in an over-the-counter product applied on a wart" is an interesting clinical case, which shows that treatment with over-the-counter products can be harmful. Patients often abuse over-the-counter products, especially natural ones, without fear of side effects. All side effects of such products should be reported and recorded. The paper contains relevant knowledge, especially for primary care physicians who may have patients with such complications. It is important to distinguish between allergic and irritant reactions. All these aspects were discussed in the article.

Reply: Thank you very much

Round 2

Reviewer 2 Report

The problem with this article that there is vast documentation and data available about tea tree oil causing contact eczema, so this particular article has NO novelty. Nothing new in it. And just one single case report!